# Vaccinia virus D10 has broad decapping activity that is regulated by mRNA splicing

**Michael Ly[1], Hannah M. Burgess[2]¤, Sahil B. Shah[3], Ian Mohr[2], Britt A. Glaunsinger[1,4,5]***

**1** Department of Molecular and Cell Biology, University of California Berkeley, Berkeley, California, United States of America, **2** Department of Microbiology, New York University School of Medicine, New York, New York, United States of America, **3** Center for Computational Biology, University of California Berkeley, Berkeley, California, United States of America, **4** Department of Plant & Microbial Biology, University of California Berkeley, Berkeley, California, United States of America, **5** Howard Hughes Medical Institute, Berkeley, California, United States of America

¤ Current address: Department of Microbial Sciences, School of Biosciences and Medicine, University of Surrey, Guildford, United Kingdom
* glaunsinger@berkeley.edu

**Data Availability Statement:** The RNA-seq data is available through the GEO accession number: GSE185520. The code used to calculate intron coordinates on the human genome is available at:

## Abstract

The mRNA 5' cap structure serves both to protect transcripts from degradation and promote their translation. Cap removal is thus an integral component of mRNA turnover that is carried out by cellular decapping enzymes, whose activity is tightly regulated and coupled to other stages of the mRNA decay pathway. The poxvirus vaccinia virus (VACV) encodes its own decapping enzymes, D9 and D10, that act on cellular and viral mRNA, but may be regulated differently than their cellular counterparts. Here, we evaluated the targeting potential of these viral enzymes using RNA sequencing from cells infected with wild-type and decapping mutant versions of VACV as well as in uninfected cells expressing D10. We found that D9 and D10 target an overlapping subset of viral transcripts but that D10 plays a dominant role in depleting the vast majority of human transcripts, although not in an indiscriminate manner. Unexpectedly, the splicing architecture of a gene influences how robustly its corresponding transcript is targeted by D10, as transcripts derived from intronless genes are less susceptible to enzymatic decapping by D10. As all VACV genes are intronless, preferential decapping of transcripts from intron-containing genes provides an unanticipated mechanism for the virus to disproportionately deplete host transcripts and remodel the infected cell transcriptome.

## Author summary

Vaccinia virus (VACV) is a DNA virus of the *poxviridae* family that was used as a vaccine for immunization against smallpox, ultimately enabling eradication of the smallpox virus. Unusual for DNA viruses, poxviruses like VACV replicate in the cytoplasm and thus must encode their own DNA replication and RNA processing machinery. This includes a protein called D10, which is a decapping enzyme that removes the protective 5' caps of messenger RNA transcripts, causing them to be degraded, which is hypothesized to decrease antiviral signaling. Here, we demonstrate that D10 targets the majority of cellular

https://github.com/SahilBShah/Vaccinia_intron_coordinates.

**Funding:** This research is funded by NIH CA136367 to BAG, NIH Grants to IM (GM056927, AI073898, AI152543) and HMB (AI151436), and NSERC Predoctoral Fellowship PGSD3-516787-2018 to ML. BAG is an investigator of and receives salary from the Howard Hughes Medical Institute. The funders had no role in study design, data collection and analysis, decision to publish, or preparation of the manuscript.

**Competing interests:** The authors have declared that no competing interests exist.

messenger RNA transcripts. However, the activity of D10 is influenced by the splicing background of a transcript, where mature transcripts that have been spliced are more targeted and degraded by D10 compared to mature transcripts that are unspliced. The ability of D10 to distinguish transcripts by their splicing history enables it to deplete human transcripts while sparing viral transcripts, reshaping the landscape in favor of viral translation.

## Introduction

The gene expression pathways for the vast majority of DNA viruses mimic those of the host cell, and they often rely on host machinery found in the nucleus for transcription, mRNA processing and splicing. A notable exception are the poxviruses such as vaccinia virus (VACV), which are unusual as DNA viruses because they replicate in the cytoplasm and therefore do not have access to the cellular mRNA synthesis and processing factors located in the nucleus [1]. This includes splicing factors, and thus in contrast to the majority of mammalian genes, all VACV genes are intronless. However, VACV encodes a sophisticated set of transcriptional and RNA processing machinery that includes a multi-subunit RNA polymerase enzyme, 5' capping complex, and 3' processing machinery [2–4]. Notably, VACV also encodes two decapping enzymes, D9 and D10, that remove the protective 5' cap on host and viral transcripts, leading to degradation of the decapped mRNAs by cellular exonucleases such as XRN1 [5,6]. The accumulation of viral double stranded RNA (dsRNA) in cells infected with VACV mutants lacking D9 and D10 suggest they contribute to removal of these innate immune-activating transcripts [7,8]. Decapping-induced degradation of host transcripts may further benefit the virus by increasing ribosome availability for viral translation.

D9 and D10 are 25% similar at the amino acid level, and while homologs of D10 are found across all poxviruses, D9 appears restricted to the chorodopoxvirus subfamily. Several observations suggest that the activity and contributions of these enzymes towards the poxvirus life-cycle are distinct. D9 activity is more sensitive to inhibition by uncapped RNA and less sensitive to inhibition by cap analogs than D10 *in vitro* [5,9,10]. Furthermore, mutagenesis of D9, which is expressed with early kinetics, has no effect on viral titers *in vivo*, while mutagenesis of D10, which is expressed late, reduces viral titers and increases expression of individually tested host and viral genes [6,11,12]. D10 has also recently been shown to enhance translation of VACV transcripts, but how its activity affects the overall abundance of the viral and host transcriptome is unclear [13,14].

In eukaryotic cells, decapping enzymes play a major role in determining mRNA fate, and thus their activity is tightly regulated [15]. The function of the cellular decapping enzyme DCP2, for example, is controlled by both *cis*-acting factors, such as autoinhibition by its C-terminal domain, and *trans*-acting proteins that regulate its activity on specific mRNA substrates [16–18]. Like DCP2, VACV D10 contains a Nudix hydrolase domain [19]; however, it lacks the inhibitory C-terminal domain, supporting the hypothesis that D10 may function as a constitutively active decapping enzyme. Indeed, many viruses encode proteins that functionally mimic cellular factors but have evolved to have reduced or novel regulation [2–4,20,21]. Here, we tested this hypothesis by defining the breadth of VACV D10 targets by RNA sequencing. We confirm that D10 is the primary contributor to mRNA downregulation during VACV infection and is a promiscuous enzyme that depletes the majority of human transcripts. Both D9 and D10 target viral transcripts for degradation in a partially redundant manner, although intermediate and late viral transcripts are less efficiently targeted by the two decapping enzymes. Despite the broad activity of D10, it does not target all transcripts comparably. We

made the surprising finding that mRNAs derived from intronless genes are relatively refractory to D10, suggesting a role for mRNA splicing in regulating D10 activity. While there are established links between viral ribonuclease substrate preference and splicing [22–24], this represents the first such connection for a viral or cellular decapping enzyme. Given that VACV genes lack introns, this also provides a mechanism for the relative protection of viral mRNAs from decapping-induced turnover.

## Results

### VACV D10 broadly reduces cellular mRNA abundance

To determine the cellular targets of the VACV D10 decapping enzyme, we first generated HEK293T stable cell lines containing N-terminally 3xFLAG-tagged wild type (WT) D10 or the D10 catalytic mutant E144Q/E145Q [9] under the control of a doxycycline (dox)-inducible promoter. Dox treatment for 18 hours resulted in comparable expression of WT and catalytically inactive D10 protein. Because the proteins were minimally expressed, they were only detectable following anti-FLAG immunoprecipitation and western blotting (Fig 1A). We confirmed by RT-qPCR that expression of WT D10, but not D10 E144Q/E145Q, caused downregulation of several housekeeping genes, including *ACTB*, *GAPDH*, and *EEF1A1* (Fig 1B). D10 was unable to target the uncapped cellular *7SK* RNA, as expected given that its activity is likely restricted to capped mRNAs.

To determine the breadth of D10 targeting, we sequenced ribodepleted total RNA from uninduced or dox-induced cells. Prior to sequencing, we added ERCC spike-in RNA into each sample to normalize for differences in library size. Each experiment was performed in three biological replicates, and there was high reproducibility between each replicate (S1A Fig). Induction of WT D10 led to broad changes in gene expression, with the majority of host transcripts being downregulated (Fig 1C), suggesting that D10 is a promiscuous decapping enzyme. In contrast, induction of D10 E144Q/E145Q led to minimal changes in gene expression (Fig 1D), consistent with the enzymatic decapping activity of D10 being required to mediate the large-scale changes in gene expression.

We next sought to determine whether the breadth of D10 targeting observed in the inducible cells similarly held true in the context of VACV infection. As VACV encodes two decapping enzymes, D9 and D10, we isolated the activity of a specific enzyme by comparing WT VACV to infection with virus in which the only decapping activity is provided by D10 (ΔD9), and where there is no decapping activity, a catalytically inactive D9 and D10 mutant (D9muD10mu). We infected HEK293T cells with each virus at an MOI of 3 and performed RNA-sequencing of three independent biological replicates at 3-, 6-, and 18-hours post-infection (hpi). Infection of select cell lines with ΔD9 and D9muD10mu leads to accumulation of dsRNA and activation of the OAS-RNase L pathway, a cell intrinsic antiviral response [8]. When activated, RNase L is an endonuclease that broadly cleaves RNAs, including ribosomal RNA (rRNA) that could confound our analysis of D10 targets. However, we did not detect hallmark rRNA cleavage products in the Bioanalyzer RNA traces from any RNA samples (S1B Fig). All biological replicates showed high reproducibility (S1C Fig). WT VACV infection did not cause large changes in gene expression at 3 and 6 hpi (S2A and S2B Fig) but did result in strong downregulation of the vast majority of host mRNAs at 18 hpi (Fig 1E). The robust downregulation of host mRNAs persisted during ΔD9 VACV infection (S2C Fig), as expected given that this virus contains functional D10. In contrast, inactivation of both D9 and D10 yielded a substantial recovery in host gene expression (Fig 1F), indicating that the decapping enzymes, and perhaps D10 catalytic activity in particular, are largely responsible for the VACV-induced depletion of cellular mRNA. Notably, viral reads accounted for 40% of the

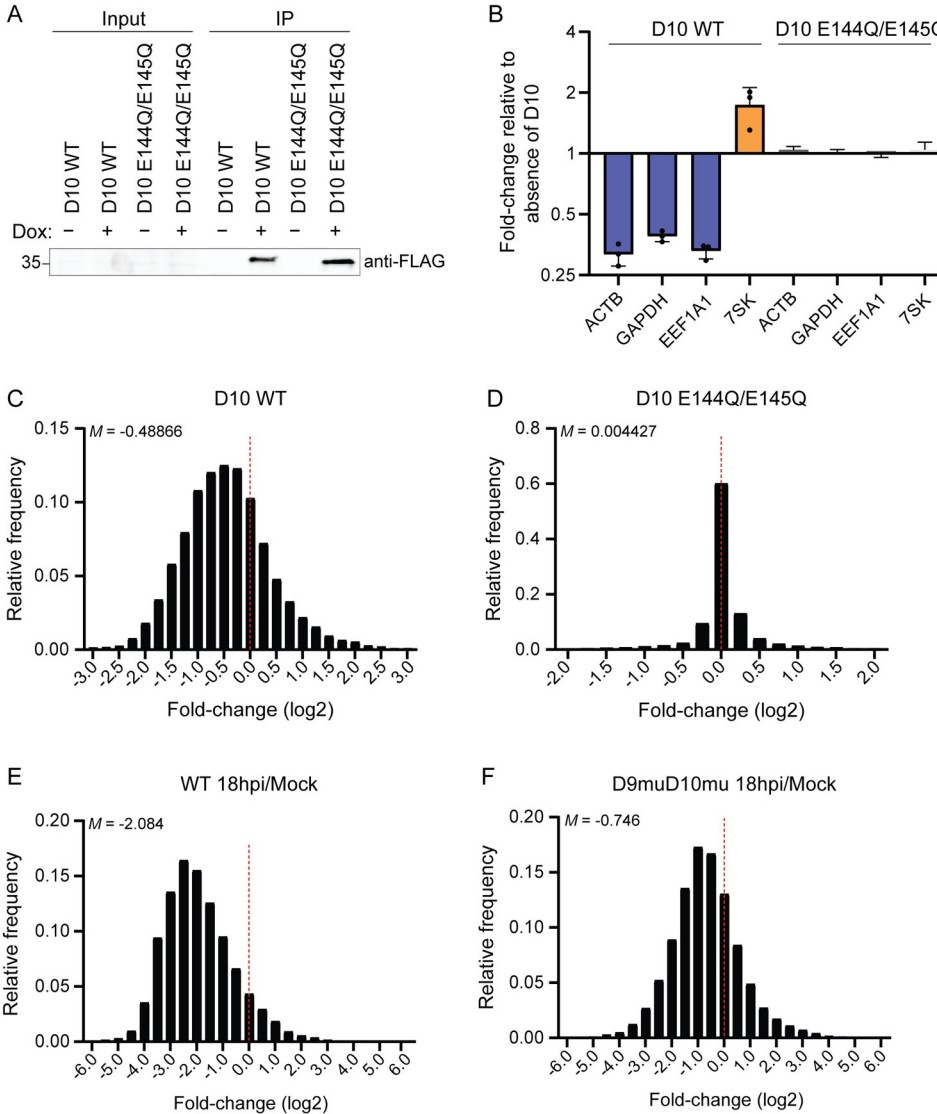

**Fig 1. VACV D10 causes global mRNA degradation.** (A) Levels of 3xFLAG-D10 WT and E144Q/E145Q were visualized by western blotting following immunoprecipitation with anti-FLAG beads. (B) RT-qPCR was used to measure levels of the indicated RNAs in either uninduced (-) or dox-induced (+) 3xFLAG-D10 WT and E144Q/E145Q HEK293T stable cells. Each point represents an independent replicate (N = 3). The bars represent the mean value of the replicates and error bars represent standard deviation. (C-D) The fold-change value for each host transcript in the 3xFLAG-D10 WT (*C*) and E144Q/E145Q (*D*) was calculated by dividing its expression in the dox-treated condition by its expression in the untreated condition. The frequency distribution for these values is plotted on each histogram. The dotted red line marks the fold-change value of zero, and the median value (*M*) is shown for each plot. (E-F) The fold-change value for each host transcript was calculated by dividing its expression at 18 hpi with either WT VACV (*E*) or D9muD10mu VACV (*F*) by its expression in mock infected cells. The frequency distribution for these values is plotted on each histogram. The dotted red line marks the fold-change value of zero, and the median value (*M*) is shown for each plot.

total reads at 18 hpi, consistent with the role of mRNA degradation in shifting the host transcriptome to one that is dominated by viral transcripts during infection (S2D Fig).

GO term enrichment analyses indicated that, in the D10 induced and WT VACV infection contexts, the most downregulated transcripts included genes involved in oxidative phosphorylation, as well as ribosomal protein genes (upon D10 induction) and mitochondrial function

(during WT VACV infection), which have been implicated in supporting viral replication [25] (S3A and S3B Fig). To examine transcripts specifically depleted by D10 during VACV infection, we isolated the activity of D10 by comparing host transcript abundance in cells infected with the ΔD9 and D9muD10mu strains at 18 hpi. GO term enrichment of this analysis revealed that the genes most robustly downregulated by D10 during infection were involved in various cellular processes, most notably apoptosis (S3C Fig). The host genes downregulated by D10 in VACV-infected cells are distinct from those repressed by inducible D10 expression in uninfected cells. Induction of apoptosis by VACV infection [26] could lead to upregulation of apoptosis-related transcripts in virus-infected cells, but not in the uninfected, D10-expressing cells.

The most upregulated genes in the D10 induced cells were those in the categories of transcription factors and RNA polymerase II, which may reflect a transcriptional response to the global depletion of cytoplasmic RNAs (S3D Fig). During infection, the most upregulated genes included those involved in signaling pathways and transmembrane channel activity (S3E Fig). Unsurprisingly, these were distinct from the categories of upregulated genes in the D10 induced cells, as many other VACV proteins likely contribute to this response.

## D10 targets transcripts derived from intronless genes less robustly compared to those from intron-containing genes

Although D10 targeted most cellular transcripts, there was large variation in the magnitude of downregulation, suggesting that RNA features may influence the sensitivity of a transcript to D10 targeting. While 5'UTR length, GC content, and folding energy did not correlate with targeting efficiency (S4A–S4C Fig), we found a moderate correlation between transcript abundance and how robustly D10 targets the transcript. This was true for D10 targeting during infection, which was measured by comparing host transcript abundance in cells infected with the ΔD9 versus the D9muD10mu strains at 18 hpi, and in the WT D10 inducible cells, suggesting that more highly expressed transcripts are more susceptible to D10 targeting (Fig 2A and 2B). Using this finding, we filtered for genes in the D10 induced cells that were highly expressed but not well targeted by D10 to search for RNA features associated with protection against D10 targeting. Among the top 15 genes in this category were several histone genes, which stood out to us because like VACV genes, these genes lack introns [27] (S4D Fig). Additionally, the genes most strongly downregulated by D10 during infection contained multiple introns, suggesting the splicing architecture of a gene may influence how well it is targeted by D10 (S4E Fig). We therefore interrogated our D10 inducible and infection-based RNA-seq data to assess whether the presence of genomic introns impact D10 targeting and found that transcripts from intron-containing genes were indeed more robustly targeted by D10 compared to those from intronless genes. This observation held true when comparing either all genes or an expression-matched dataset of genes within the 50th to 75th percentile of RNA abundance and upon removal of uncapped noncoding RNAs (e.g., snoRNAs, which are intronless) from the dataset (Figs 2C, 2D and S5A–S5E).

Over the full range of splicing events per mRNA, we did not observe a negative correlation with RNA destabilization in the D10-inducible cells, suggesting that increasing number of splicing events does not proportionally sensitize a transcript for stronger degradation mediated by D10 (S6A Fig). However, we examined this more rigorously by assessing the effects of D10 induction on groups of transcripts with various exon numbers to determine if a singular or multiple splicing events influence D10 targeting efficiency. Since there were expression differences between the groups (S6B Fig), we analyzed the expression-matched dataset and observed similar D10 targeting of transcripts from genes with one intron (i.e., 2 exons) and

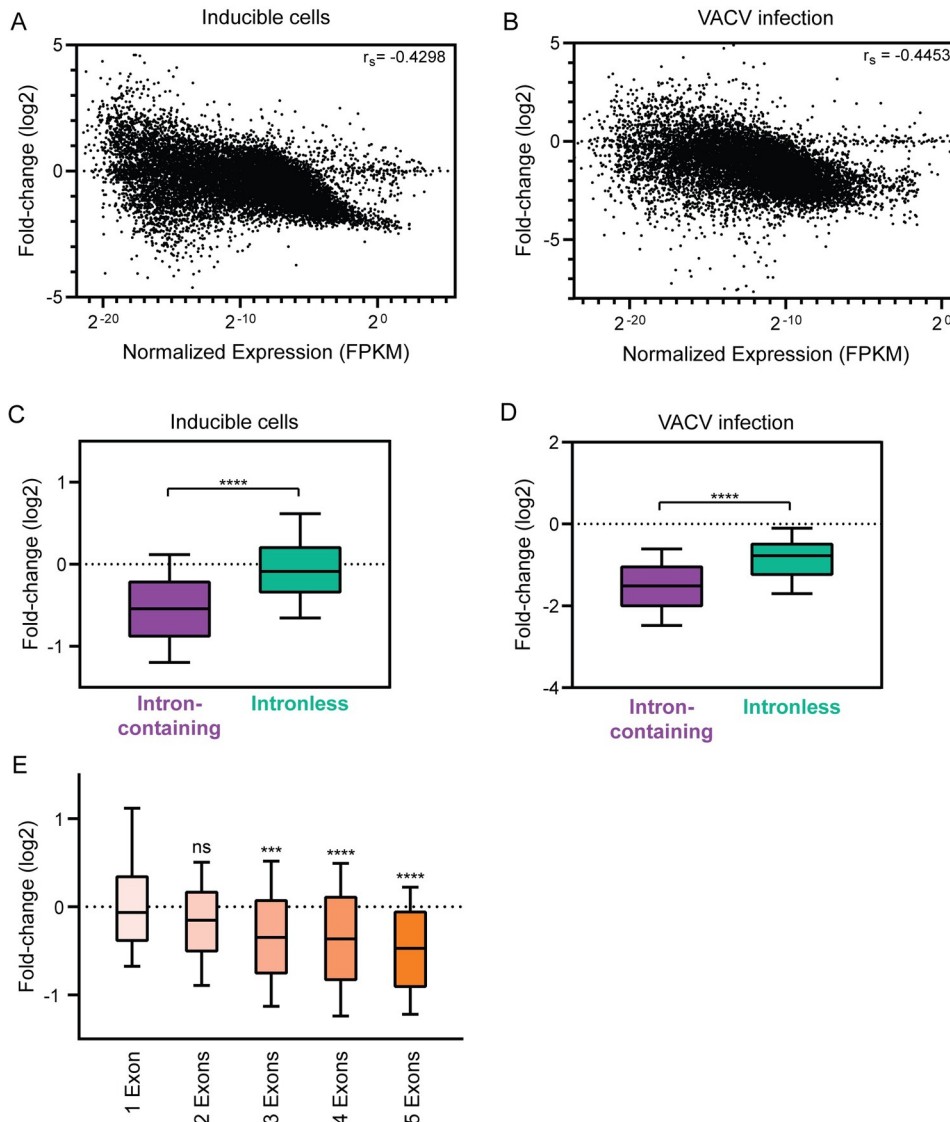

**Fig 2. Intron-containing genes are more robustly downregulated by D10 than intronless genes.** (A) The fold-change value upon WT D10 induction was correlated against the expression level of each host transcript. The Spearman correlation coefficient ($r_s$) is shown. Out of 17,704 data points, there are 108 data points outside the axis limits. (B) To isolate the contribution of D10 during infection, the fold-change value for each host transcript was calculated by dividing its expression in cells infected with ΔD9 VACV by its expression in D9muD10mu VACV at 18 hpi. These values were correlated against the expression level of each transcript. The Spearman correlation coefficient ($r_s$) is shown. Out of 14,630 data points, there are 201 data points outside the axis limits. (C-D) Cellular transcripts, excluding ncRNAs, within expression-matched dataset were divided into intron-containing or intronless category and plotted against their fold-change value reflecting D10 activity in dox-inducible cells (*C*) or during infection (*D*). ****P≤0.0001, Mann-Whitney test. The whiskers represent the 10th and 90th percentiles. (E) Cellular transcripts within expression-matched dataset were binned according to the number of exons they contain and plotted against their fold-change value upon D10 induction. ns-not significant, ***P≤0.001, ****P≤0.0001, Kruskal-Wallis test followed by Dunn's multiple comparison test versus "1 Exon". The whiskers represent the 10th and 90th percentiles.

transcripts from intronless genes (i.e., 1 exon) (Fig 2E). In contrast, genes with two or more introns (i.e., 3 exons or more) were targeted more robustly by D10, suggesting that multiple splicing events may be required to increase a transcript's sensitivity to D10.

## Spliced transcripts are more sensitive to D10 activity

We validated the above observation by measuring the D10 sensitivity of several expression-matched intron-containing (*ERCC8*, *GBE1*, *GJA1*) and intronless (*DDX28*, *CCDC8*, *ZNF830*) (S2 Table) genes by RT-qPCR, which similarly showed that transcripts from intronless genes were refractory to D10 activity (Fig 3A). Furthermore, D10 did not appear to target its own mRNA (derived from an intronless cDNA) in the D10-inducible, uninfected cells, as the level of *D10* mRNA in WT D10 expressing cells was even higher than in the cells expressing the catalytically inactive mutant D10 E144Q/E145Q (Fig 3B). Altogether, these data are consistent with the hypothesis that transcripts derived from intronless genes are less sensitive to D10 activity.

As transcripts with multiple exons could be longer than those with a single exon, we considered the possibility that transcript length could be contributing to the differential activity exerted by D10. However, there was no significant correlation between transcript length and D10 targeting efficiency (S6C Fig). To directly test if splicing sensitizes a transcript to D10, we compared the ability of D10 to target two *IFNL2* mRNAs of identical sequence that differed only in whether they were transcribed from an intronless cDNA version or a 5 intron-containing genomic version of the reporter plasmid [22]. We controlled for potential differences in transfection efficiency by including plasmids encoding either adenovirus (Adv)-derived VAI RNA or B2 SINE RNA, both of which are uncapped Pol III transcripts and thus not susceptible to decapping by D10. In these co-transfection assays, D10 depleted the genomic DNA-derived *IFNL2* mRNA significantly more robustly than the cDNA-derived *IFNL2* mRNA (Figs 3C and S6D). The reduction in the abundance of *IFNL2* mRNA can be attributed to cytoplasmic mRNA degradation by D10, as its levels were partially rescued in the absence of the cellular 5'-3' exonuclease Xrn1 [28] (Fig 3D). Altogether, these results are consistent with the role of splicing in sensitizing transcripts to decapping by D10.

Next, we evaluated D10 localization by fractionating the 3xFLAG-D10-inducible cells into nuclear and cytoplasmic fractions and immunoprecipitating D10 to monitor its expression in each compartment. D10 was present in both the cytoplasm and the nucleus, whereas GAPDH and histone H3 were restricted to their expected cytoplasmic or nuclear compartments, respectively (S6E Fig). Since D10 is present in the nuclear fraction, we investigated whether pre-mRNAs might be targeted by D10 before intron removal as a mechanism to distinguish transcripts derived from intron-containing and intronless genes. Reads mapping to introns are an accurate readout of pre-mRNA levels [29]. While exonic reads were reduced upon WT D10 expression, intronic reads were unchanged (Fig 3E), suggesting that D10 does not target pre-mRNAs in the nucleus. In addition, we reasoned that intron-containing genes might possibly be longer and take more time to be fully transcribed, and thus their transcripts are retained in the nucleus for longer and subjected to increased D10 activity. However, we did not detect a correlation between gene length and D10 targeting efficiency (S6F Fig), arguing against transcriptional kinetics being the primary factor for the susceptibility differences we observe between transcripts derived from intron-containing and intronless genes. Thus, although D10 is partially localized to the nucleus, its preferential decapping of transcripts from intron-containing genes does not appear to occur co-transcriptionally.

## D9 and D10 target an overlapping set of viral transcripts but to a lesser extent than host transcripts

We next assessed viral transcript targeting by comparing WT to ΔD9 VACV infection datasets to identify D9-specific changes and ΔD9 to D9muD10mu VACV infection datasets to identify D10-specific changes. Most viral transcripts were modestly upregulated with the loss of D9 or

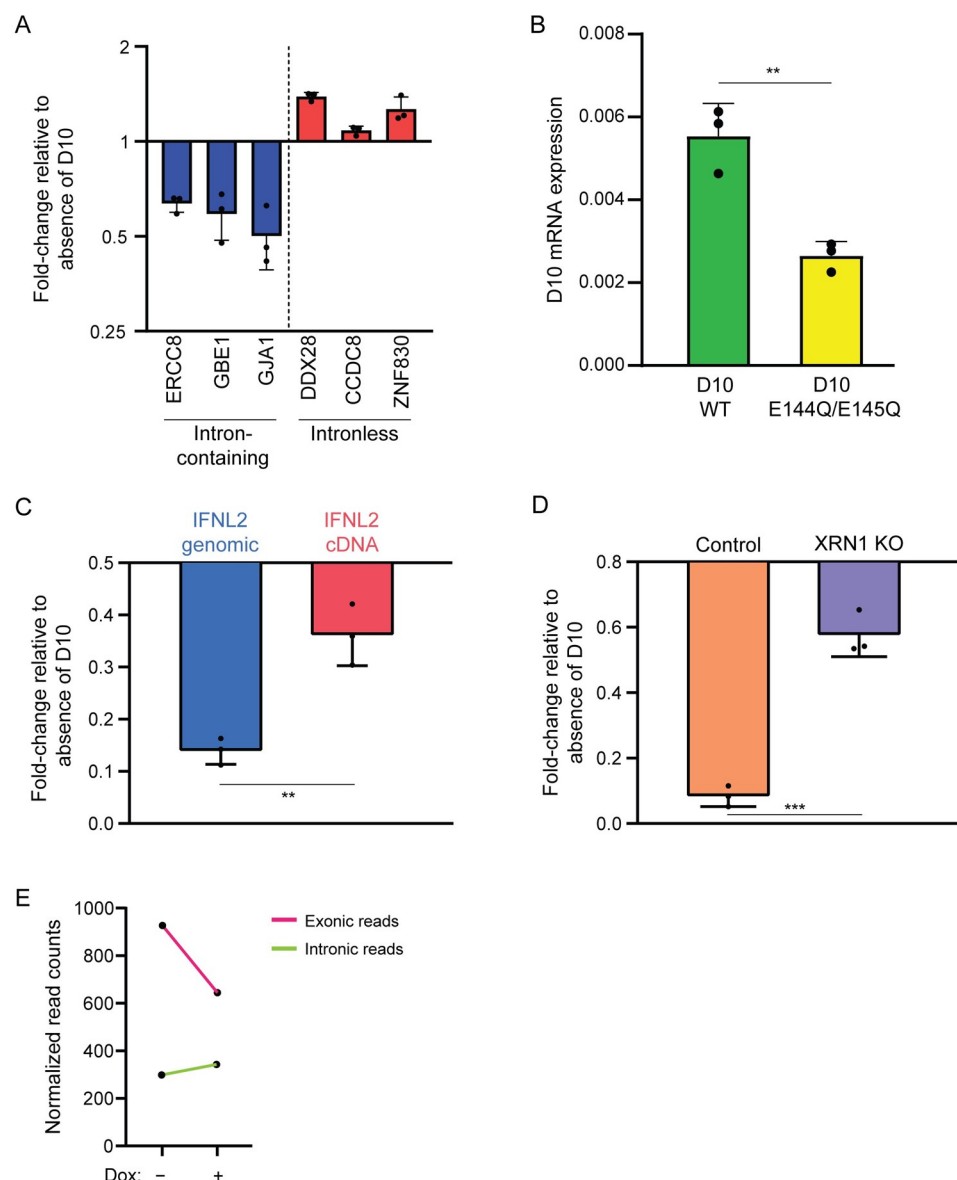

**Fig 3. Splicing sensitizes transcripts for stronger decapping by D10.** (A) 3xFLAG-D10 WT HEK293T stable cells were either uninduced and dox-induced, and RT-qPCR was used to calculate fold-change of the indicated transcripts in the dox-induced samples relative to uninduced. Each point represents an independent replicate (N = 3). The bars represent the mean value of the replicates and error bars represent standard deviation. (B) 3xFLAG-D10 WT and E144Q/E145Q HEK293T stable cells were dox-induced, and RT-qPCR was used to assess D10 mRNA abundance relative to 18S. Each point represents an independent replicate (N = 3). **P≤0.01, unpaired t-test. The bars represent the mean value of the replicates and error bars represent standard deviation. (C) HEK293T cells were co-transfected with D10, Adv-VAI, and either the genomic (5 introns) or intronless cDNA version of the IFNL2 reporter. RT-qPCR was used to quantify levels of IFNL2, which were normalized to Adv-VAI RNA, and the fold-change was calculated relative to absence of D10. Each point represents an independent replicate (N = 3). **P≤0.01, unpaired t-test. The bars represent the mean value of the replicates and error bars represent standard deviation. (D) Control or XRN1 knockout (KO) HEK293T cells were co-transfected with D10, Adv-VAI, and genomic version of IFNL2 reporter. RT-qPCR was used to quantify levels of IFNL2, which were normalized to Adv-VAI RNA, and the fold-change was calculated relative to absence of D10. Each point represents an independent replicate (N = 3). ***P≤0.001, unpaired t-test. The bars represent the mean value of the replicates and error bars represent standard deviation. (E) The normalized total exonic or intronic read counts from either uninduced (-) or dox-induced (+) 3xFLAG-D10 WT HEK293T stable cells.

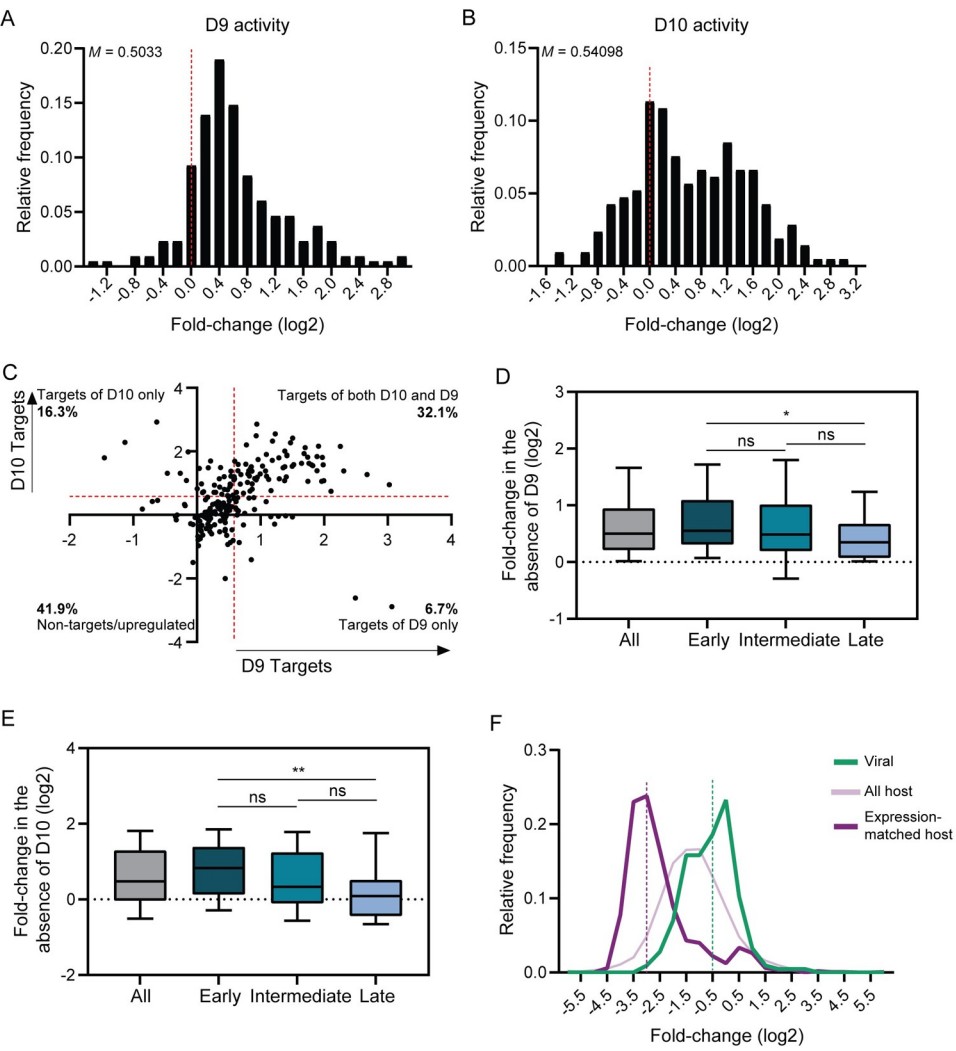

**Fig 4. D10 preferentially targets host over viral transcripts.** (A) To isolate the activity of D9 on viral transcripts, the fold-change value for each viral transcript was calculated by dividing its expression in cells infected with ΔD9 VACV by its expression in WT VACV at 18 hpi. The histogram shows the frequency distribution for these values, with the dotted red line marking the fold-change value of zero. The median value (*M*) is shown for the plot. (B) To isolate the activity of D10 on viral transcripts, the fold-change value for each viral transcript was calculated by dividing its expression in cells infected with D9muD10mu VACV by its expression in ΔD9 at 18 hpi. The histogram shows the frequency distribution for these values, with the dotted red line marking the fold-change value of zero. The median value (*M*) is shown for the plot. (C) The values calculated from *(A)* and *(B)* were plotted against each other. The targets of D9 and D10 were assigned as transcripts upregulated by at least 1.5-fold (0.585 on log2 scale; red line) with the loss of D9 or D10, respectively. The percentage of viral mRNAs falling into each quadrant is shown. (D-E) Viral transcripts from each kinetic class (early, intermediate, late) or in aggregate (all) were plotted against their fold-change in the absence of D9 (*D*) or D10 (*E*). *P≤0.05, **P≤0.01, Kruskal-Wallis test followed by Dunn's multiple comparison test versus "Early" genes. The whiskers represent the 10th and 90th percentiles. (F) The activity of D10 towards host and viral transcripts was assessed using a similar approach as (*B*), except taking the reciprocal to make the fold-change values negative. A subset of expression-matched host transcripts, defined as highly expressed transcripts within the 90th-100th percentile of expression, was also analyzed. The histogram shows the frequency distribution for these values, with the dotted purple and green lines representing the bin where the median value of downregulation lies for expression-matched host and viral genes, respectively.

D10, and loss of both enzymes resulted in upregulation by at least 1.5-fold of an overlapping set of 32.1% of VACV mRNAs (Fig 4A–4C). However, some transcripts were selectively targeted by D9 (6.7%) or D10 (16.3%), and 41.9% of viral transcripts escaped decay by D9 and

D10 (Fig 4C and S1 Table). Notably, among the kinetic classes of viral genes, late transcripts were least efficiently targeted (Fig 4D–4E). We therefore hypothesized that late transcript expression would not be significantly altered in the D9muD10mu virus, where both D9 and D10 are inactivated, compared to WT virus. Indeed, the abundance of early transcripts was significantly upregulated in the D9muD10mu virus whereas the abundance of intermediate and late transcripts was not affected (S7A Fig).

Given that D10 preferentially targets spliced mRNAs and all VACV genes are intronless, we compared the extent of D10 targeting of host and viral transcripts. Indeed, the magnitude of downregulation elicited by D10 on host transcripts was much greater than on viral transcripts (median values 2.34-fold vs 1.45-fold, respectively) (Fig 4F). Furthermore, considering that D10 targets highly abundant transcripts more robustly and viral transcripts are more abundant than host transcripts (S7B Fig), we also compared its targeting to a subset of expression-matched host transcripts, defined as highly expressed transcripts within the 90th-100th percentile of expression. In this expression-matched comparison, the difference in D10 targeting of host and viral transcripts was even larger (median values 7.31-fold vs 1.45-fold) (Fig 4F). Thus, while D10 promotes widespread depletion of cellular mRNA, its activity on viral transcripts is tempered, presumably by the absence of viral mRNA splicing and perhaps other features inherent to late VACV transcripts.

## Discussion

Many viruses encode "host shutoff" factors that promote widespread depletion of cellular mRNA, both to enhance viral access to translation machinery and as a general immune evasion mechanism. However, an emerging theme is that these seemingly promiscuous viral enzymes have properties that confer selectivity and influence targeting efficiency. Understanding the mechanisms involved in the mRNA targeting of these viral enzymes provides insight into cellular reprogramming by a spectrum of viruses. Here, we probed the target selectivity of the VACV decapping enzymes, revealing how the activities of both D9 and D10 contribute to shaping RNA abundance during infection. While D9 primarily targets viral transcripts for decay, D10 depletes a much broader repertoire of mRNAs, with stronger activity against transcripts derived from intron-containing genes than those derived from intronless genes. This is the first example of a decapping enzyme whose substrate preference appears influenced by splicing, which suggests an unanticipated connection between these seemingly distinct gene regulatory processes. It also provides a mechanism for sensitizing host over viral mRNA decapping. The robust activity against host transcripts could liberate ribosomes for viral protein synthesis, as D10 has been reported to promote translation of reporters containing the 5' poly (A) tract, a feature found on intermediate and late viral transcripts [14,30–32]. Additionally, the more limited degradation of viral mRNAs could reduce the accumulation of immune activating viral dsRNAs without significantly compromising viral translation.

Given that D9 and D10 are expressed with distinct temporal kinetics during infection, it has been hypothesized that they may target similar transcripts but at different time points during infection [8,12]. Our comparative data with mutant VACV infections indicate that there does appear to be redundant decapping of viral transcripts, although there are subsets of viral transcripts uniquely targeted by only one of the enzymes (e.g., 6.7% are targeted by only D9). The targeting of viral transcripts is modest relative to host transcripts, and a large fraction of viral transcripts escape decay by both enzymes. Unlike viral transcripts, decapping of host transcripts appears to be predominantly executed by D10, as we detected no difference in cellular mRNA abundance between the WT vs. ΔD9 VACV infections. Furthermore, depletion of host mRNAs was not prominent at early stages of infection when D9, but not D10, is

expressed. However, it remains possible that the presence of D10 compensates for loss of D9 activity on cellular mRNA and thus future experiments focused on D9 targeting may better reveal its host targeting potential.

Although D10 targets the majority of cellular mRNA, it does so with unequal efficiency as our data suggest that multiple splicing events sensitize transcripts to D10-mediated decapping. This provides a potential mechanism to protect viral transcripts against robust decapping, as they are all intronless whereas the vast majority of mammalian mRNAs are derived from intron-containing genes. It is also notable that late viral transcripts were less efficiently targeted by D10 compared to early viral transcripts. One distinguishing feature of viral transcripts expressed after viral DNA replication is the presence of a long poly(A) tract at their 5' end [30–32], and previous *in vitro* characterization of D10 showed reduced decapping efficiency on reporters with adenosine as their first nucleotide [9]. Both the lack of splicing and the presence of a 5' poly(A) tract on late transcripts presumably contribute to the relative protection of viral transcripts against promiscuous decapping. Although it has not been demonstrated that the length of the 5' poly(A) tract influences D9 or D10 targeting, it would be valuable to comprehensively measure 5' poly(A) tract length of viral transcripts and establish the extent to which it impacts D9 and D10 targeting. Additionally, a recent study showed that D10 is localized to the mitochondria, which are excluded from the viral factories where viral transcripts are synthesized [33]. The spatial regulation of D10 may therefore further contribute to its preferential activity on host transcripts over viral transcripts. This observation is also interesting considering our finding that the most downregulated transcripts by D10 in inducible, uninfected cells included genes involved in oxidative phosphorylation. In contrast, the most downregulated transcripts by D10 during infection were involved in apoptosis and other cellular pathways. The difference between these two contexts may be because infected cells have increased levels of apoptosis-related transcripts, as VACV is known to induce apoptosis [26]. The targeting of apoptosis-related transcripts by D10 could alter the host-virus dynamics and may be beneficial for the virus; for example, by delaying apoptosis to produce more progeny virions.

A key open question is how D10 discriminates between transcripts derived from intron-containing vs intronless genes. One possibility is that D10 recruitment is enhanced by factors co-transcriptionally deposited on transcripts during splicing. Leading candidates for this model are components of exon-junction complexes (EJCs), which are deposited at exon-exon junctions after splicing and participate in several aspects of RNA metabolism as well as promote degradation of transcripts bearing premature termination codons [34]. If EJCs influence D10 recruitment to mRNA, they would have to do so prior to the pioneer round of translation as they are removed by translating ribosomes. Another possibility is that D10 interacts with the spliceosome machinery and is itself co-transcriptionally loaded onto mRNAs. D10 is present both in the cytoplasm and nucleus, raising the possibility that it could target mRNA in both compartments. Notably, based on intronic read counts, we detected minimal changes in pre-mRNA levels upon D10 expression. This suggests that, if D10 is co-transcriptionally loaded onto mRNAs through the spliceosome, its targeting would likely occur after splicing and transcription are completed.

Our findings are consistent with the role of mRNA splicing in regulating the enzymatic decapping activity of D10, an unexpected mode of regulation that has not been described for any other decapping enzyme. Nonetheless, the concept of splicing as a mechanism to direct preferential depletion of host transcripts is emerging as a common theme in viruses that restrict cellular gene expression. This has been described for the influenza A endoribonuclease PA-X, which interacts with factors associated with the spliceosome complex and stimulates nuclear degradation of mRNA derived from intron-containing genes [22]. Recently, splicing has also been shown to sensitize transcripts for stronger degradation by the SARS-CoV-2 nsp1 protein,

a pathogenicity factor that has a dual function of inhibiting translation and eliciting mRNA cleavage [23]. Because genes from VACV, influenza A, and SARS-CoV-2 are largely intronless, the preferential targeting of mRNAs from intron-containing genes by otherwise broad acting viral proteins likely contributes to viral transcript escape and preferential translation. The reverse example of preferentially targeting unspliced mRNAs, during the period when spliced mRNAs are transiently insensitive due to EJC binding, can also occur, as shown for the herpes simplex virus (HSV) UL41 endoribonuclease [24]. Taken together, these results suggest that splicing or the splicing history of an mRNA comprise a determinant of virus-induced RNA decay triggered by virus-encoded endoribonucleases or decapping enzymes.

Interestingly, a large fraction of human intronless genes are enriched for growth factors and play roles in translation or energy metabolism [35]. It might therefore be beneficial for the virus to spare these transcripts, as their replication depends on host resources. Conversely, all mammalian type I interferon genes are intronless [36], which could be an evolutionary response from the host to protect innate immunity transcripts from viral depletion.

## Materials and methods

### Plasmids and cloning

3xFLAG-D10 WT was cloned through an InFusion cloning reaction (Takara Bio) into the NotI site of pCDNA4 to generate pCDNA4-3xFLAG-D10 WT. Single primer-based mutagenesis was used to generate the pCDNA4-3xFLAG-D10 E144Q/E145Q mutant. The doxycycline-regulated promoter-containing pLVX-3xFLAG-D10 WT and E144Q/E145Q vectors for lentivirus production were constructed by an InFusion cloning reaction. The IFNL2-genomic and IFNL2-cDNA reporters were gifts from Dr. Marta Gaglia (Tufts University) [22]. The expression vectors for B2 SINEs and Adv-VAI were obtained from a prior study in the lab [37].

### Cell culture, lentiviral transduction, and transfections

All cell lines used for viral titration, propagation, or infection (HEK293T, BSC40, BHK-21, and RK13) were grown in Dulbecco's modified Eagle's medium (Corning) supplemented with 10% fetal bovine serum (Gibco) and 1% penicillin/streptomycin (v/v) at 37°C and 5% $CO_2$. For the dox-inducible cell line, HEK293T cells were grown in Dulbecco's modified Eagle's medium (Gibco) supplemented with 10% fetal bovine serum (Peak Serum) at 37°C and 5% $CO_2$. To produce lentivirus and generate stable cells, $2.5\times10^6$ HEK293T cells were seeded into each well of a 6-well plate. The next day, they were transfected with lentiviral packaging vectors using TransIT-LTI transfection reagent (Mirus Bio) according to the manufacturer's protocol. Specifically, 1250 ng psPAX2, 250 ng pCMV-VSVG, and 1250 ng pLVX-3x-FLAG-D10 WT or pLVX-3xFLAG-D10 E144Q/E145Q were used for transfection. After 48 hours, the lentivirus-containing supernatant was collected and suspended with $2.5\times10^6$ HEK293T cells. The cell-virus mixture was then centrifuged at 2,000 rpm, 37°C for 2 hours with 2 μg/mL polybrene. The HEK293T cells were subsequently transferred to a 10 cm dish and selected with 350 μg/mL zeocin for one week. For doxycycline induction experiments, cells were treated with 1 μg/mL doxycycline for 18 hours prior to sample collection. For IFNL2 co-transfection experiments, $1\times10^6$ HEK293T cells were seeded into each well of a 6-well plate. The next day, they were transfected with 900 ng Adv-VAI or B2 SINEs, 100 ng pCDNA4-D10, and 1 ng pCMV-IFNL2-genomic or pCMV-IFNL2-cDNA using PolyJet transfection reagent (SignaGen labs) according to the manufacturer's DNA transfection protocol. The cells were harvested 24 hours post-transfection for analysis. The control and XRN1 knockout HEK293T cells were generated from a previous study [28].

## Viral infection

Wild-type VacV (Western Reserve strain) and ΔD9 viruses were propagated and titered in BSC40 cells, as described previously [7]. Double mutant virus vD9muD10mu was propagated in BHK-21 cells and titered in RK13 cells. All mutant viruses were kindly supplied by Dr. Bernard Moss (NIH NIAID, Bethesda). For the RNA-seq experiment, HEK293T cells were infected at an MOI of 3, and lysates were collected in TRIzol at the relevant timepoints post infection.

## FLAG immunoprecipitation and western blot analysis

Cells were lysed with lysis buffer (50 mM Tris pH 7.4, 150 mM NaCl, 1 mM EDTA, 0.5% NP-40) supplemented with protease inhibitor tablet (Roche) and rotated for 30 minutes at 4˚C. The cell lysate was then spun for 10 minutes at 15,000 rpm, 4˚C. For cell fractionations, we performed REAP fractionation as described previously [38]. For each immunoprecipitation, 2.5 mg of protein was mixed with 20 μL of anti-FLAG M2 magnetic bead slurry (Sigma) in a 1 mL volume, using lysis buffer without detergent to fill the volume. The mixture was rotated overnight at 4˚C and subsequently washed three times with wash buffer (50 mM Tris pH 7.4, 150 mM NaCl, 1 mM EDTA, 0.05% NP-40) supplemented with protease inhibitor tablet (Roche), with 5 minutes of rotation at 4˚C per wash. The beads were resuspended in 25 μL 2X Laemmli buffer and boiled to elute immunoprecipitants (IP). For western blot analysis, 1% input and 100% of IP were used for SDS-PAGE and probed with anti-FLAG antibody (1:3000, Sigma F7425).

## RNA extraction and RT-qPCR

Cells were lysed in TRIzol reagent (ThermoFisher), and RNA was extracted following the manufacturer's protocol. The RNA was then treated with TURBO DNase (ThermoFisher), and subsequently subjected to reverse transcription using AMV Reverse Transcripase (Promega) with a random 9-mer primer. cDNA was then used in qPCR following the iTaq Universal SYBR Green Supermix protocol (Bio-Rad laboratories) and gene-specific qPCR primers (S3 Table).

## RNA sequencing

The RNA was isolated from TRIzol samples using Direct-zol RNA MiniPrep Plus (Zymo) following the manufacturer's protocol and subsequently treated with TURBO DNase (ThermoFisher). The RNA was purified using RNA Clean & Concentrator Kit (Zymo), and to normalize for differences in library size, ERCC RNA Spike-In Control Mix (ThermoFisher) was added into each sample following the manufacturer's protocol. The integrity of the samples was analyzed using Fragment Analyzer Service provided by QB3 at UC Berkeley. The samples were sent to QB3 for ribodepletion and library generation. After multiplexing the samples, they were sequenced on the HiSeq 4000 (for dox-induction experiment) or NovaSeq 6000 (for VACV infection experiment) to obtain paired-end 150 nucleotide reads.

## Read alignment and bioinformatic analysis

Paired reads were trimmed using TrimGalore v0.4.4 at default settings except—stringency 3. The UCSC human genome (hg19) and ERCC spike-in sequences were combined and indexed using STAR v2.7.1a. For viral sequence alignment, the VACV genome (NC_006998.1) was also combined and indexed. Reads were mapped onto the genomes using STAR v2.7.1a. To calculate the Spearman correlation coefficients between biological replicates, the FPKM value

for each replicate was computed using CuffLinks v2.2.1 at default settings except -library-type and fr-firststrand -p 20. All genes with FPKM value of 0 were removed. Genes from all replicates were matched, and Spearman correlation coefficient was calculated between each replicate in a pairwise manner. For differential gene expression analysis, CuffDiff (CuffLinks v2.2.1) was used to compare two conditions (e.g., untreated versus dox-induced), using all three biological replicates from each condition. The default settings for CuffDiff were mostly used except -library-type fr-firststrand, -library-norm-method classic-fpkm, and -p 20. All genes with FPKM value of 0 were removed, and for genes with multiple transcript isoforms, only the most highly expressed transcript isoform was used for downstream analysis. This was done because there was a high degree of variability amongst lowly expressed transcripts. Prior to calculating fold-change, the FPKM value for each transcript was normalized to the sum of ERCC spike-in FPKM values from each condition to control for differences in library size. For analysis of exonic and intronic read counts, HTSeq v0.9.1 was used to count the reads mapping to exonic or intronic regions of genes. The default settings were used except -f bam -r pos -s reverse -t intron (or exon), and -i gene_id. To obtain the genomic coordinates for introns, we used an in-house script to calculate the coordinates between adjacent exons of all genes. The sum of exonic reads or intronic reads were then normalized to the sum of reads mapping to the ERCC spike-ins from each condition. Exon number information was obtained from UCSC Human Gene Sorter. 5'UTR information was obtained from Ensembl Genome Browser. R Studio was used throughout to merge files.

### GO analysis

GO term analysis was done using Gene Ontology enRIchment anaLysis and visuaLizAtion tool (GOrilla). A rank list of genes was submitted, and the program searched for enriched GO terms that appear densely at the top of the list. Molecular Function ontology was selected for, and the P-value threshold was set at $10^{-3}$. The results were then entered into REViGO to coalesce related GO terms and summarize the findings.

### Data visualization

All data plots and graphs were generated using GraphPad Prism v9.1.2 and edited using Adobe Illustrator for color coordination and aesthetic purposes. Tables and graphs depicting GO analysis were generated in Microsoft Excel and edited using Adobe Illustrator for aesthetic purposes.

### Statistical analyses

All statistical analyses were performed in GraphPad Prism v9.1.2 using the tests indicated in the figure legends.

## Supporting information

**S1 Fig. RNA-seq of dox-inducible D10 stable cells and VACV-infected cells.** (A) Spearman correlation coefficient between biological replicates from RNA-seq experiment in dox-inducible cells. (B) Bioanalyzer trace results of the RNA samples from VACV-infected HEK293T cells. (C) Spearman correlation coefficient between biological replicates from RNA-seq in VACV-infected HEK293T cells.
(TIF)

**S2 Fig. RNA degradation does not occur early on in infection.** (A-B) The fold-change value for each host transcript was calculated by dividing its expression in WT VACV by its

expression in mock infected cells at 3 hpi (*A*) or 6 hpi (*B*). The frequency distribution for these values is plotted on each histogram. The dotted red line marks the fold-change value of zero, and the median value (*M*) is shown for each plot. (C) The fold-change value for each host transcript was calculated by dividing its expression in ΔD9 VACV by its expression in mock infected cells at 18 hpi. The frequency distribution for these values is plotted on the histogram. The dotted red line marks the fold-change value of zero, and the median value (*M*) is shown for each plot. (D) Percentage of reads mapping to the viral or host genome during WT VACV infection at 3, 6, or 18 hpi.
(TIF)

**S3 Fig. GO term analysis of genes in D10 inducible cells and during VACV infection.** (A-C) GO term analysis for Molecular Function ontology among ranked list of genes downregulated by D10 in dox-inducible cells (*A*), downregulated by WT VACV infection at 18 hpi relative to mock (*B*), or downregulated specifically by D10 during VACV infection at 18 hpi (*C*). Based on their P-values, the top 10 enriched GO terms are shown. (D-E) GO term analysis for Molecular Function ontology among ranked list of genes upregulated by D10 in dox-inducible cells (*D*) or upregulated by WT VACV infection at 18 hpi relative to mock (*E*). Based on their P-values, the top 10 enriched GO terms are shown.
(TIF)

**S4 Fig. RNA features that influence D10 targeting.** (A-C) Correlation between 5'UTR GC % (*A*), length (*B*), or folding energy (*C*) and fold-change of cellular transcripts upon D10 induction. The Spearman correlation coefficient ($r_s$) is shown. For (*A*) and (*B*), there are 78 data points outside the axis limits out of a total of 14,967 data points. For (C), there are 40 data points outside the axis limits out of a total of 12,357 data points. (D) Cellular transcripts were sorted by high expression level then sequentially by low fold-change value upon D10 induction, with the table showing the top 15 genes in this category. (E) Table of top 15 genes most downregulated by D10 during infection, reflected by comparing host transcript abundance in cells infected with the ΔD9 and D9muD10mu strains at 18 hpi, and their corresponding number of introns.
(TIF)

**S5 Fig. Relationship between splicing and D10 targeting.** (A) Cellular transcripts were divided into intron-containing or intronless category and plotted against their fold-change value upon D10 induction. ****P≤0.0001, Mann-Whitney test. The whiskers represent the 10th and 90th percentiles. (B) Transcript abundance of intron-containing and intronless genes from (*A*) was compared. ****P≤0.0001, Mann-Whitney test. The dotted red line represents the median value. (C) Transcript abundance of expression-matched dataset, which consists only the subset of genes expressed within 50th to 75th percentile range. ns-not significant, Mann-Whitney test. The dotted red line represents the median value. (D-E) Cellular transcripts from expression-matched dataset were divided into intron-containing or intronless category and plotted against their fold-change value reflecting D10 activity in dox-inducible cells (*D*) or during infection (*E*). ****P≤0.0001, Mann-Whitney test. The whiskers represent the 10th and 90th percentiles.
(TIF)

**S6 Fig. Low correlation between other RNA features and D10 targeting.** (A) Correlation between exon number and fold-change of cellular transcripts upon D10 induction. The Spearman correlation coefficient ($r_s$) is shown. Out of 13,841 data points, there are 36 data points outside the axis limits. (B) Transcript abundance of cellular transcripts binned according to the number of exons they contain. ns-not significant, *P≤0.05, ***P≤0.001, ****P≤0.0001,

Kruskal-Wallis test followed by Dunn's multiple comparison test versus "1 Exon". The dotted red line represents the median value. (C) Correlation between transcript length and fold-change of cellular transcripts upon D10 induction. The Spearman correlation coefficient ($r_s$) is shown. There are 271 data points outside the axis limits out of a total of 15,593 data points. (D) HEK293T cells were co-transfected with D10, B2 SINE, and either the genomic (5 introns) or intronless cDNA version of the IFNL2 reporter. RT-qPCR was used to quantify levels of IFNL2, which were normalized to B2 SINE, and the fold-change was calculated relative to absence of D10. Each point represents an independent replicate (N = 3). $^*$P$\leq$0.05, unpaired t-test. The bars represent the mean value of the replicates and error bars represent standard deviation. (E) 3xFLAG-D10 WT HEK293T stable cells were either uninduced (-) or dox-induced (+) then fractionated into nuclear and cytoplasmic compartments. The levels of 3xFLAG-D10 in each compartment were visualized by western blotting following immunoprecipitation with anti-FLAG beads, while the levels of Histone H3 (nuclear marker) and GAPDH (cytoplasmic marker) in each compartment were detected in the input samples. (F) Correlation between gene length and fold-change of cellular transcripts upon D10 induction. The Spearman correlation coefficient ($r_s$) is shown. There are 81 data points outside the axis limits out of a total of 15,573 data points.
(TIF)

**S7 Fig. Late transcripts are the least susceptible amongst viral transcripts to targeting by D9 and D10.** (A) The abundance of viral transcripts from different kinetic classes was compared between WT VACV and D9muD10mu VACV at 18 hpi. ns-not significant, $^{***}$P$\leq$0.001, $^{****}$P$\leq$0.0001, Kruskal-Wallis test followed by Dunn's multiple comparison test versus "WT VACV" infection. The whiskers represent the 10th and 90th percentiles. (B) Transcript abundance of viral transcripts, host transcripts, and a subset of expression-matched host transcripts, defined as highly expressed transcripts within the 90th-100th percentile of expression. ns-not significant, $^{****}$P$\leq$0.0001, Kruskal-Wallis test followed by Dunn's multiple comparison test versus "Viral". The dotted red line represents the median value.
(TIF)

**S1 Table. Viral transcripts targeted by D9 and/or D10.**
(XLSX)

**S2 Table. Number of introns found in the endogenous genes quantified by qPCR in this study**
(XLSX)

**S3 Table. Oligos used in this study.**
(XLSX)

## Acknowledgments

We thank the labs of Marta Gaglia and Bernard Moss for providing the reagents used in this study. We appreciate members of the Glaunsinger Lab (Azra Lari, Valeria King, Aaron S Mendez) for their constructive feedback on this manuscript.

## Author Contributions

**Conceptualization:** Michael Ly, Hannah M. Burgess, Ian Mohr, Britt A. Glaunsinger.

**Data curation:** Sahil B. Shah.

**Formal analysis:** Michael Ly, Hannah M. Burgess, Sahil B. Shah.

**Funding acquisition:** Ian Mohr, Britt A. Glaunsinger.

**Investigation:** Michael Ly, Hannah M. Burgess.

**Methodology:** Michael Ly.

**Resources:** Sahil B. Shah.

**Supervision:** Ian Mohr, Britt A. Glaunsinger.

**Validation:** Michael Ly.

**Writing – original draft:** Michael Ly, Britt A. Glaunsinger.

**Writing – review & editing:** Michael Ly, Hannah M. Burgess, Ian Mohr, Britt A. Glaunsinger.

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
