## [Decision Letter · Decision Letter 0]

20 Dec 2021

Dear Dr. Glaunsinger,

Thank you very much for submitting your manuscript "Vaccinia virus D10 has broad decapping activity that is regulated by mRNA splicing" for consideration at PLOS Pathogens. As with all papers reviewed by the journal, your manuscript was reviewed by members of the editorial board and by several independent reviewers. The reviewers appreciated the attention to an important topic. Based on the reviews, we are likely to accept this manuscript for publication, providing that you modify the manuscript according to the review recommendations.

The three reviewers were enthusiastic regarding your submission and found the intron-containing preference for the vaccinia decapping enzyme to be interesting, novel and very impactful. All of the reviewers suggested additional revisions to support the key conclusions of the study as well as reveal some additional details of this novel mechanism. I believe that these suggestions are valuable, generally addressable, and will provide valuable validation for the key conclusions of the study. Thus I encourage you to strongly consider them in a revised manuscript.

Sincerely,

Jeffrey Wilusz

Guest Editor

PLOS Pathogens

Klaus Früh

Section Editor

PLOS Pathogens

Kasturi Haldar

Editor-in-Chief

PLOS Pathogens

orcid.org/0000-0001-5065-158X

Michael Malim

Editor-in-Chief

PLOS Pathogens

orcid.org/0000-0002-7699-2064

The three reviewers were enthusiastic regarding your submission and found the intron-containing preference for the vaccinia decapping enzyme to be interesting, novel and very impactful. All of the reviewers suggested additional revisions to support the key conclusions of the study as well as reveal some additional details of this novel mechanism. I believe that these suggestions are valuable, generally addressable, and will provide valuable validation for the key conclusions of the study. Thus I encourage you to strongly consider them in a revised manuscript.

Reviewer Comments (if any, and for reference):

Reviewer's Responses to Questions

**Part I - Summary**

Reviewer #1: Many viruses possess the capacity to manipulate the 5’ end of host and viral mRNAs to facilitate propagation. The Vaccinia virus (VACV) encode two proteins, D9 and D10, that function to remove the 5’ protective m7G cap (decapping) from mRNA. Decapping and degradation of host cell mRNAs is an important strategy to minimize translation from host-cell encoded mRNAs and enhance viral protein translation. In this manuscript the Glaunsinger lab utilizes RNA-seq from Wt VACV,D9 and/or D10 mutant VACA virus infected mammalian cells to identify viral and host cell RNAs targeted by the D9 and D10 decapping proteins. D9 and D10 both target a subset of viral RNAs at a different stage of viral infection. The authors uncover a very interesting observation that D10 predominantly targets host cell mRNAs encoded by intron-containing genes providing a mechanism to preferentially deplete predominantly intron containing host mRNAs while retaining the intronless viral transcripts. This is a significant finding that helps explain the selective advantage that D10 decapping could provide the virus and possibly new insight into the regulation of cellular decapping enzymes. The following are several suggests that would improve the manuscript.

Reviewer #2: Ly and colleagues examined the role for the vaccina virus D10 decapping enzyme in the viral infectious cycle. It was found that D10 decaps both viral and cellular transcripts. However, D10 prefers intron-containing transcripts, allowing the intron-less viral transcripts to escape from degradation. The results are supported by excellent genetic and computational studies that reveal a novel link between decapping and splicing in modulating virus-host interactions.

Reviewer #3: In this work, Li et al. examine the effect of viral VACV D10 mRNA decapping enzyme on cellular and viral transcripts using RNA-Seq analysis. The authors use two different approaches to examine the effect of D10 on cellular transcripts: 1) through inducible expression of VACV D10 in human cells and 2) during VACV infection. The data show that D10 has a profound effect on cellular mRNA levels, particularly on mRNAs derived from genes containing introns, a feature notably absent from VACV genes. The authors also present data regarding the effect of VACV D10 and D9 on viral transcript levels.

Overall, the paper presents a novel and intriguing intron-based model for viral mRNA decapping enzymes to preferentially target host transcripts as opposed to viral mRNAs. The specific targeting of host mRNAs could promote viral mRNA access to the ribosomes, along with the shutdown of host protein synthesis.

**Part II – Major Issues: Key Experiments Required for Acceptance**

Reviewer #1: (No Response)

Reviewer #2: 1. D10 is expressed late in viral infection. Why would decapping of cellular transcripts be beneficial for viral gene expression when, for example, ribosomes are needed early during infection?

2. Does D10 locate to the nucleus?

3. The authors measure RNA abundances throughout their studies and link them to degradation. Could it be that transcription is affected as well by D10? Studies in XRN1-depleted cells could clarify this point.

Reviewer #3: 1. The authors provide evidence that mRNAs expressed from genes containing introns are preferentially targeted by VACV D10 for decapping and subsequent degradation. Since this is a novel mechanism for viral-mediated host mRNA decapping, additional supporting evidence would be helpful. Furthermore, since VACV is a cytoplasmic virus, and D10 localization is primarily cytoplasmic/mitochondrial, it is hard to envision a model in which cellular cytoplasmic, spliced mRNAs could be targeted once the exon junction complexes are removed. Of course, this does not preclude the possibility of splicing influencing VACV mRNA decapping, but additional evidence would make the claim more convincing.

One way this could be accomplished is through discussion of specific mRNAs most targeted by VACV D10 and their corresponding gene structure. For example, similar to Figure S4C, a table could be provided indicating the mRNAs most affected by D10, along with the corresponding intron architecture of the gene encoding it. These data would be particularly interesting in the context of VACV infection. Furthermore, for Figure 1B, it would be helpful to know the number of introns for ACTB, GAPDH, EEF1A1, and the same would be useful for Figure 3A for ERCC8, GBE1, and GJA1. (Side note: It is fascinating the mRNAs targeted most by VACV function in the mitochondria and during oxidative phosphorylation.)

A second way this could be accomplished would be analysis of additional mRNA species in Figures 3A, C, and D, to ensure that another mRNA feature is not responsible for the targeting by VACV D10.

Lastly, the authors examined whether GC content or 5’ UTR length influenced targeting by VACV D10. Other mRNA features could also be ruled out using the RNA-Seq data, such as sequence elements in the 5’ or 3’ UTR. This seems particularly important since cellular Dcp2 preferentially recognizes mRNAs with a 5’ stem-loop structure.

2. The dataset describing the difference in viral transcripts in the D9/D10 mutant backgrounds is extremely useful. It would be helpful if more analysis were performed on this dataset to make the data more meaningful, such as the GO Term analysis performed for host genes, along with analysis of 5’ poly(A) leader structure of the viral mRNAs in relation to D9/D10 targeting. Also, it would have been useful to analyze the �D10 mutant virus to account for effects of D9 alone on viral transcripts, but this experiment could be performed in the future.

**Part III – Minor Issues: Editorial and Data Presentation Modifications**

Reviewer #1: 1. The authors postulate two reasonable mechanisms for the preferential targeting of D10 to introned transcripts in the discussion section. One being a deposition of D10 onto the premRNA by the spliceosome in the nucleus and a second being a recruitment to spliced mRNA through an interaction of D10 with the exon junction complex in the cytoplasm. Although definitively answering how D10 targets intron-containing transcripts is beyond this manuscript, some additional suggestive data would significantly strengthen it. For example, is there any evidence for D10 interactions in protein interactome databases? Does the RNA seq data reveal whether the D10 transcripts were primarily pre-mRNA or mRNA (providing information of what compartment the regulation may occur).

In addition, have the authors considered a kinetic explanation? Does the susceptibility to D10 correlate with pre-mRNA size where longer transcripts (ie multi intron containing transcripts) are retained in the nucleus longer than shorter intronless transcripts and more susceptible to D10?

2. Line 191: A 1.2 fold cutoff is used to evaluate percentage of different transcript that are responsive to D9 or D10. What is the rationale for using 1.2 rather than the more common cutoffs of 2 fold or even 1.5 fold?

Reviewer #2: Fig. 1A. There is no input signal.

Reviewer #3: Introduction:

1. Line 53: Also increased levels of certain viral mRNAs…

2. Line 62: I suggest addition of semi-colon after “however”.

3. Line 71: I suggest Despite “the” broad activity of D10.

Results

4. Throughout the results sections, it would be helpful if authors clearly defined the cell line or mutant strain used for each experiment: e.g., 3XFLAG-D10 or 3XFLAG-D10 (E144Q/E145Q) or �D9 compared to vD9muD10mu. I also suggest adding these labels to the figures in the paper for increased sample clarity.

5. Line 87: Figure 1B Given later findings, what is exon/intron number for these housekeeping genes?

6. Lines 88-89: Figure 1B Why is 7SK RNA increased when D10 is expressed?

7. Lines 143-145: What is exon/intron structure of other genes besides histone genes (e.g., RPL37A)?

8. Line 154: What is exon/intron structure of ERCC, GBE1, etc.?

Figures

9. Figure 1: Suggest adding labels for C and D cell line D10 protein version expressed (3XFLAG-D10 and 3XFLAG-D10 (E144Q/E145Q) and for E and F mutant strain (e.g., �D9 for E) and, hours post-infection—this would be useful for all subsequent figures as mentioned above.

PLOS authors have the option to publish the peer review history of their article (what does this mean?). If published, this will include your full peer review and any attached files.

Reviewer #1: No

Reviewer #2: No

Reviewer #3: No

Figure Files:

Data Requirements:

Reproducibility:

References:

---

## [Editor Report · Decision Letter 1]

10 Feb 2022

Dear Dr. Glaunsinger,

We are pleased to inform you that your manuscript 'Vaccinia virus D10 has broad decapping activity that is regulated by mRNA splicing' has been provisionally accepted for publication in PLOS Pathogens.

Best regards,

Jeffrey Wilusz

Guest Editor

PLOS Pathogens

Klaus Früh

Section Editor

PLOS Pathogens

Kasturi Haldar

Editor-in-Chief

PLOS Pathogens

orcid.org/0000-0001-5065-158X

Michael Malim

Editor-in-Chief

PLOS Pathogens

orcid.org/0000-0002-7699-2064

The authors have done an excellent job in responding to the points made in the original round of critiques. I find the manuscript to be improved, convincing and impactful.
---

## [Editor Report · Acceptance letter]

20 Feb 2022

Dear Dr. Glaunsinger,

We are delighted to inform you that your manuscript, "Vaccinia virus D10 has broad decapping activity that is regulated by mRNA splicing," has been formally accepted for publication in PLOS Pathogens.

Best regards,

Kasturi Haldar

Editor-in-Chief

PLOS Pathogens

orcid.org/0000-0001-5065-158X

Michael Malim

Editor-in-Chief

PLOS Pathogens

orcid.org/0000-0002-7699-2064